# AN EMPIRICAL STUDY ON PROMPT COMPRESSION FOR LARGE LANGUAGE MODELS

**Zheng Zhang[1], Jinyi Li[2], Yihuai Lan[1], Xiang Wang[3], Hao Wang[1]**[*]
[1]The Hong Kong University of Science and Technology (Guangzhou)
[2]South China University of Technology
[3]University of Science and Technology of China
`zzhang302@connect.hkust-gz.edu.cn, haowang@hkust-gz.edu.cn`

## ABSTRACT

Prompt engineering enables Large Language Models (LLMs) to perform a variety of tasks. However, lengthy prompts significantly increase computational complexity and economic costs. To address this issue, we study six prompt compression methods for LLMs, aiming to reduce prompt length while maintaining LLM response quality. In this paper, we present a comprehensive analysis covering aspects such as generation performance, model hallucinations, efficacy in multimodal tasks, word omission analysis, and more. We evaluate these methods across 13 datasets, including news, scientific articles, commonsense QA, math QA, long-context QA, and VQA datasets. Our experiments reveal that prompt compression has a greater impact on LLM performance in long contexts compared to short ones. In the Longbench evaluation, moderate compression even enhances LLM performance. Our code and data is available at `https://github.com/3DAgentWorld/Toolkit-for-Prompt-Compression`.

## 1 INTRODUCTION

Large Language Models (LLMs) have demonstrated remarkable generalization capabilities (Grosse et al., 2023; Yang et al., 2024), allowing them to adapt to a wide range of tasks through prompt engineering techniques such as CoT (Wei et al., 2024), ICL (Dong et al., 2024), and RAG (Lewis et al., 2020) without necessitating fine-tuning. However, this advantage comes with an obvious drawback: increasing the length of prompts to encompass the necessary information, which subsequently escalates computational overhead (Wang et al., 2024). Also, for online models such as ChatGPT and Claude, lengthy prompts inflate the economic cost associated with API calls.

To address this issue, prompt compression is the most straightforward strategy. As illustrated in Figure 1, it aims to reduce the length of prompts while retaining the essential information. However, previous works (Li et al., 2023; Jiang et al., 2024; Pan et al., 2024) have primarily focused on how LLMs perform on various tasks (e.g. summarization, reconstruction and question answering) using common metrics (e.g. accuracy, BLEU (Papineni et al., 2002b), ROUGE (Lin, 2004b) and BERTScore (Devlin et al., 2019)) after applying prompt compression. There has been a noticeable gap in understanding how prompt compression affects other aspects of LLM output, beyond the specific task performance.

Specifically, the effects on aspects such as generalizability and hallucinations have not been thoroughly examined. Moreover, existing works rarely apply prompt compression to Multimodal LLMs (MLLMs), raising questions about the generalizability of compression techniques in multimodal tasks. Furthermore, what kind of prompt words can be omitted when prompting is also under-investigated. This may provide valuable insights for more effective prompt engineering strategies.

Therefore, it is crucial to explore the broader impacts of different prompt compression methods on (M)LLMs across different tasks.

---

[*]Corresponding author.

In this paper, we address these issues by conducting comprehensive studies with three (M)LLMs (GPT-3.5-turbo, GPT-4o-mini, Claude-3-Haiku) on 13 datasets, including news, scientific articles, common sense QA, math QA, long context QA, and VQA datasets.

Technically, we design our empirical study to address the following questions: **(1)** Which prompt compression method performs best across different tasks? How does compression ratio affect performance? **(2)** Does prompt compression affect other aspects of the models output, such as response length and hallucinations? **(3)** Are current prompt compression approaches generally effective when applied to MLLMs for multimodal tasks? **(4)** What kind of words can be omitted when prompting?

Our **key findings** can be summarized as follows:

- (Long)LLMLingua and LLMLingua-2 generally outperform other methods, especially at high compression ratios.
- All methods' performance decreases with increasing compression ratios for short contexts, but for long contexts, moderate compression can improve performance.
- Prompt compression can influence response length, with the direction of change depending on the specific LLM.
- All methods result in some degree of increased hallucination, with information loss being the primary reason.

Figure 1: **Illustration of prompt compression.** The original context is distilled into a more concise form while preserving pertinent information for LLMs to process. Some methods compress the context based on the query, while others do not. Words that are underlined in the original text denote the segments that are trimmed by the compressor.

Our contributions can be summarized as follows: **(1)** We present a comprehensive study that evaluates various prompt compression methods across different tasks. **(2)** By analyzing the effects of prompt compression on response length, hallucinations, and its generalizability in multimodal contexts, we provide insights beyond traditional metrics. **(3)** We compile our implementation into an open-source toolkit, facilitating further research in prompt compression for LLMs.

## 2 RELATED WORKS

### 2.1 LLM'S LONG CONTEXT PROCESSING METHOD

Given the performance limitations and computational overhead of LLMs (Wang et al., 2024), how to effectively apply LLMs to tasks involving lengthy textual inputs is a persistent challenge. Various solutions have emerged to address this issue, encompassing techniques such as length extrapolation (Chen et al., 2021; Shaw et al., 2018), attention approximation (Winata et al., 2019; Wang et al., 2020), attention-free transformers (Gu et al., 2021; Orvieto et al., 2023), model compression (Lee et al., 2023; Ma et al., 2023), and hardware-aware transformers (Dao et al., 2022; Liu & Abbeel, 2023).

In this paper, we focus mainly on the prompt compression techniques, especially those that do not rely on the internal states or parameters of LLMs and operate in a text-in, text-out manner. These methods present several advantages: they can be seamlessly integrated with different model architectures without requiring additional modifications, and they are particularly beneficial for online models, helping to reduce the economic costs associated with API calls.

### 2.2 PROMPT COMPRESSION

Figure 1 illustrates the concept of prompt compression, and the compression ratio $\rho$ for prompt compression is defined as:

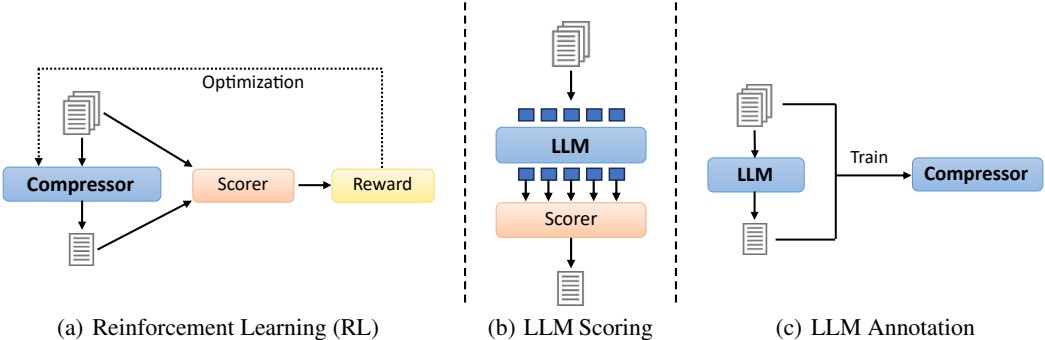

(a) Reinforcement Learning (RL)    (b) LLM Scoring    (c) LLM Annotation

Figure 2: **Categories of prompt compression methods.** These methods can be grouped into three main categories: (a) RL-based methods, which use heuristic rewards to optimize the compressor, (b) LLM scoring-based methods, which use another language model to score each token in a single autoregressive step and decide to keep or discard each token based on its score, and (c) LLM annotation-based methods, which use LLMs to annotate data for training a small model specifically designed for prompt compression.

$$\rho = 1 - \frac{L_c}{L_o}. \tag{1}$$

Here $L_c$ is the compressed context length and $L_o$ is the original context length. Many prompt compression methods have been developed to handle long prompts in LLMs. KiS (Laban et al., 2021) and SCRL (Ghalandari et al., 2022) leverage reinforcement learning (RL) to train models for text compression without the need for ground-truth data, optimizing specific objectives such as fluency and simplicity. Recently, with advances in LLMs, some methods (Li et al., 2023; Jiang et al., 2023; 2024; Pan et al., 2024) employ pre-trained language models and various strategies to identify and prune redundant or less informative content.

Besides text-based methods, there are techniques aimed at compressing or trimming the hidden states or KV caches (Liu et al., 2023b; Zhang et al., 2023; Xiao et al., 2024; Ge et al., 2024). However, these methods are separate from our study and are not easily applicable to various model architectures or closed-source LLMs.

## 3 METHODS

Figure 2 illustrates the workflows for three categories of prompt compression methods, from which we select six methods: (1) *RL-based:* KiS, SCRL, (2) *LLM scoring-based:* Selective Context, and (3) *LLM annotation-based:* LLMLingua, LongLLMLingua, LLMLingua-2. Among them, KiS does not typically trim words but uses an autoregressive approach to regenerate a shorter context, which can be time-intensive. However, we include it for comparison.

**KiS.** Laban et al. (2021) tackles the challenge of text simplification in an unsupervised manner, balancing fluency, salience, and simplicity. The model leverages reinforcement learning to enhance its performance by generating multiple candidate simplifications and optimizing for a composite reward. Utilizing a k-SCST algorithm, KiS generates $k$ candidate outputs for each input, computing a reward for each, and promotes candidates surpassing the mean reward.

**SCRL.** Ghalandari et al. (2022) also represents unsupervised sentence compression via reinforcement learning, focusing on sequence labeling. It fine-tunes a pre-trained transformer model using a simple policy gradient approach. Each token in a sentence is labeled as essential or non-essential, optimizing the reward functions to maximize the compression quality while maintaining fluency and faithfulness.

**Selective Context.** Li et al. (2023) involves assessing the informativeness of lexical units by computing their self-information using a base causal language model. By pruning the redundant parts, a more concise context is obtained.

**LLMLingua.** Jiang et al. (2023) introduces a coarse-to-fine prompt compression method to handle lengthy prompts. LLMLingua includes a budget controller to ensure semantic integrity during high compression ratios, a token-level iterative compression algorithm to model interdependencies, and instruction tuning to align distributions between a small model and LLMs.

**LongLLMLingua.** Building on LLMLingua, LongLLMLingua (Jiang et al., 2024) is tailored for long context scenarios. It employs a question-aware coarse-to-fine compression technique and re-orders documents to mitigate position bias (Liu et al., 2023a). It supports dynamic compression ratios and includes a post-compression strategy to ensure the preservation of content integrity.

**LLMLingua-2.** Developed as an advancement over LLMLingua, LLMLingua-2 (Pan et al., 2024) focuses on task-agnostic prompt compression for enhanced generalizability and efficiency. It introduces a data distillation procedure from GPT-4, creating an extractive text compression dataset to align with compression objectives effectively. LLMLingua-2 frames prompt compression as a token classification task using a Transformer encoder to leverage full bidirectional context, addressing the reliance on unidirectional context in prior approaches.

# 4 EXPERIMENT SETUP

## 4.1 TASKS AND DATASETS

For our study on prompt compression for LLMs, we designated three tasks: summarization, reconstruction, and question answering (QA). The summarization task involves generating summaries from both the original and compressed contexts and measuring the similarity between these summaries. We use datasets including Gigaword (Rush et al., 2015), DUC2004 (Over et al., 2007), BNC (Consortium, 2007), Google (Filippova & Altun, 2013), and Broadcast (Clarke & Lapata, 2008a). The reconstruction task involves prompting the LLM to reconstruct the original prompt from the compressed prompt and includes datasets like GSM8K (Cobbe et al., 2021), BBC News, Arxiv articles, and ShareGPT (Li et al., 2023). The QA task[1] leverages datasets including LongBench (Bai et al., 2024), BBH (Suzgun et al., 2023), and GSM8K[2].

For MLLMs, our primary focus is on their performance in the VQA task, utilizing datasets including IconQA (Lu et al., 2021) and OK-VQA (Marino et al., 2019). Further details about these datasets can be found in Appendix A.1.

## 4.2 METRICS

For the summarization and reconstruction tasks, we utilize BLEU[3] Papineni et al. (2002b), ROUGE[4] (Lin, 2004b), and BERTScore[5] (Zhang* et al., 2020) to measure the similarity between the generated and reference outputs. For QA and VQA tasks, we differentiate the evaluation metrics based on the nature of the answers. For tasks with clear, precise answers, accuracy is used as the evaluation metric. For open-ended questions, we assess the similarity between the generated responses and reference answers using F1 (Bai et al., 2024). For hallucination detection, following (Li et al., 2024), we use micro hallucination rate (MiHR) and macro hallucination rate (MaHR) to evaluate the degree of hallucination. Further details about the computation of these metrics can be found in Appendix A.2.

## 4.3 IMPLEMENTATIONS

In our experiments, we selected the six prompt compression methods mentioned in Section 3: KiS, SCRL, Selective Context, LLMLingua, LongLLMLingua, and LLMLingua-2. For KiS and SCRL, the compression ratio is self-adapted, while for Selective Context, LLMLingua, LongLLMLingua, and LLMLingua-2, the compression ratio is adjustable. We set it to 0.5 unless otherwise specified.

---

[1]We also categorize mathematical problems and multiple-choice questions under the scope of QA.

[2]We utilized GSM8K in both reconstruction and QA tasks. For the former, we only evaluate the performance of reconstruction without providing answers.

[3]https://github.com/nltk/nltk

[4]https://github.com/pltrdy/rouge

[5]https://github.com/Tiiiger/bert_score

Table 1: **Performance for reconstruction and summarization tasks.** We grouped LLMLingua and LongLLMLingua together, as the performance differences between these two compressors were minimal for these two tasks. For each setting, we averaged the scores across three models: GPT-3.5-turbo, GPT-4o-mini, and Claude-3-Haiku.

| Method | Metric | Reconstruction | | | | Summarization | | | | |
|---|---|---|---|---|---|---|---|---|---|---|
| | | GSM8K | BBC News | ShareGPT | Arxiv | Gigaword | DUC2004 | BNC | Broadcast | Google |
| Random Selection | | 0.40 | 0.23 | 0.19 | 0.07 | 0.25 | 0.21 | 0.21 | 0.10 | 0.23 |
| KiS | | 0.58 | 0.16 | 0.11 | 0.05 | 0.20 | 0.23 | 0.55 | 0.49 | 0.36 |
| SCRL | | 0.38 | 0.14 | 0.30 | 0.10 | 0.28 | 0.28 | 0.57 | 0.47 | 0.46 |
| Selective Context | BLEU (↑) | 0.58 | **0.33** | **0.35** | 0.31 | 0.26 | 0.25 | 0.56 | 0.50 | 0.47 |
| (Long)LLMLingua | | **0.76** | 0.19 | 0.26 | 0.15 | 0.22 | 0.23 | **0.66** | **0.73** | 0.42 |
| LLMLingua-2 | | 0.55 | 0.28 | 0.26 | **0.45** | **0.29** | **0.34** | 0.46 | 0.61 | **0.48** |
| Random Selection | | 0.60 | 0.43 | 0.54 | 0.22 | 0.21 | 0.15 | 0.32 | 0.44 | 0.37 |
| KiS | | 0.75 | 0.32 | 0.38 | 0.13 | 0.20 | 0.18 | 0.63 | 0.61 | 0.49 |
| SCRL | | 0.55 | 0.30 | 0.61 | 0.28 | 0.22 | 0.15 | 0.44 | 0.43 | 0.40 |
| Selective Context | ROUGE L (↑) | 0.82 | **0.67** | **0.66** | **0.56** | 0.21 | 0.15 | 0.59 | 0.59 | 0.54 |
| (Long)LLMLingua | | **0.89** | 0.52 | 0.58 | 0.45 | 0.23 | 0.19 | **0.80** | **0.88** | **0.56** |
| LLMLingua-2 | | 0.86 | 0.47 | 0.48 | 0.35 | **0.26** | **0.21** | 0.64 | 0.57 | 0.53 |
| Random Selection | | 0.93 | 0.85 | 0.87 | 0.81 | 0.83 | 0.84 | 0.84 | 0.82 | 0.85 |
| KiS | | 0.94 | 0.88 | 0.86 | 0.82 | 0.85 | **0.87** | 0.92 | 0.91 | 0.92 |
| SCRL | | 0.69 | 0.84 | 0.89 | 0.85 | 0.84 | 0.84 | 0.85 | 0.82 | 0.88 |
| Selective Context | BERTScore (↑) | 0.96 | **0.90** | **0.91** | **0.92** | 0.85 | 0.85 | 0.89 | 0.89 | 0.89 |
| (Long)LLMLingua | | **0.98** | 0.87 | 0.90 | 0.89 | 0.85 | **0.87** | **0.95** | **0.96** | **0.93** |
| LLMLingua-2 | | 0.94 | 0.85 | 0.86 | 0.84 | **0.86** | 0.85 | 0.90 | 0.91 | 0.90 |

Table 2: **Performance for QA tasks.** For each setting, we averaged the scores across three models: GPT-3.5-turbo, GPT-4o-mini, and Claude-3-Haiku.

| Method | BBH Boolean Expression | BBH Causal Judgement | BBH Web of Lies | GSM8K Math | LongBench SingleDoc | LongBench MultiDoc | LongBench FewShot | LongBench Synth. |
|---|---|---|---|---|---|---|---|---|
| | Acc. (↑) | Acc. (↑) | Acc. (↑) | Acc. (↑) | F1 (↑) | F1 (↑) | Acc. (↑) | Acc. (↑) |
| Original Prompt | 0.516 | 0.648 | 0.556 | 0.337 | 0.149 | 0.095 | 0.334 | 0.174 |
| Random Selection | 0.468 | 0.556 | 0.532 | 0.030 | 0.146 | 0.108 | 0.356 | 0.192 |
| KiS | **0.576** | 0.480 | 0.512 | 0.149 | 0.118 | 0.092 | 0.312 | 0.166 |
| SCRL | 0.464 | 0.472 | **0.556** | 0.218 | 0.214 | 0.302 | 0.378 | 0.176 |
| Selective Context | 0.480 | **0.616** | 0.552 | 0.179 | 0.185 | 0.101 | 0.412 | 0.288 |
| LLMLingua | 0.528 | 0.504 | 0.536 | **0.297** | 0.286 | 0.319 | 0.620 | 0.128 |
| LongLLMLingua | 0.524 | 0.536 | 0.492 | 0.218 | **0.301** | **0.334** | **0.640** | **0.582** |
| LLMLingua-2 | 0.484 | 0.584 | 0.520 | 0.220 | 0.223 | 0.312 | 0.632 | 0.210 |

Additionally, we included a random selection strategy, which involves randomly picking words from the original prompt based on the compression ratio, to serve as a baseline comparison. We evaluated these methods' performance across three (M)LLMs: GPT-3.5-turbo, GPT-4o-mini, and Claude-3-Haiku.

Moreover, we have compiled our implementation into a comprehensive toolkit, which we have open-sourced to facilitate reproducibility and further research. More details about the toolkit are provided in Appendix C.

# 5 EXPERIMENTAL RESULTS

## 5.1 MAIN RESULTS

**Question 1:** *Which prompt compression method performs best across different tasks?*

Table 1 presents a detailed comparison of different prompt compression methods, assessing their performance in various scenarios. Summarization tasks focus on retaining critical information, while reconstruction tasks emphasize detail preservation. Table 2 and Figure 4 are utilized to elucidate the performance of these methods in QA tasks with varying context lengths, from shorter contexts (BBH, GSM8K) to longer ones (LongBench). Furthermore, we assess the computational overhead of these methods, as provided in Table 3, to determine their practicality concerning time cost and memory consumption.

Table 3: **Computational overhead for different prompt compression methods.** "Time per token" refers to the time taken divided by the number of tokens removed. All metrics are evaluated on a single A6000 GPU with 48 GB memory.

| Method | Time per Prompt (ms) | Time per Token (ms) | Memory (MB) |
|---|---|---|---|
| KiS | 2410 | 5.03 | 1378 |
| SCRL | **67** | **0.15** | **315** |
| Selective Context | 319 | 0.69 | 487 |
| LLMLingua | 180 | 0.39 | 5309 |
| LongLLMLingua | 184 | 0.40 | 5309 |
| LLMLingua-2 | 115 | 0.25 | 2137 |

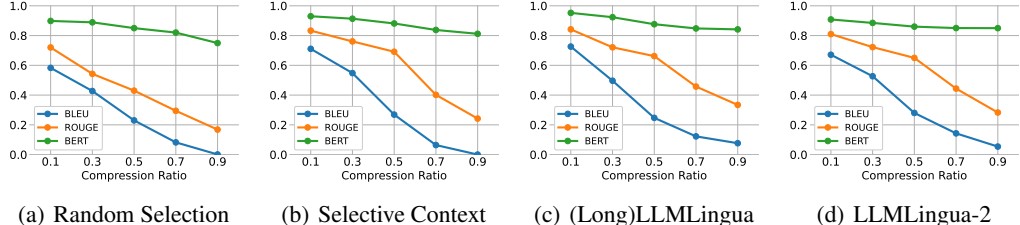

(a) Random Selection     (b) Selective Context     (c) (Long)LLMLingua     (d) LLMLingua-2

Figure 3: **Performance on compression tasks under different compression ratios.** We measured the performance of four compression methods by changing the compression ratio while keeping all other settings in accordance with Table 1. For each dataset, we randomly sampled 100 instances for evaluation and averaged their metrics. As mentioned in Section 4.3, KiS and SCRL cannot adjust the compression ratio and are thus not considered.

Our main findings are the following:

- *(Long)LLMLingua and LLMLingua-2 excel in summarization tasks, while Selective Context leads in reconstruction tasks.* (Long)LLMLingua is best for math contexts (GSM8K), LLMLingua-2 for news articles (Gigaword, DUC2004), and Selective Context for human-centric datasets (BBC News, ShareGPT). We observed that (Long)LLMLingua and LLMLingua-2 retain tokens that are concentrated around semantically rich sections of the text, which helps in creating summaries that capture the essential points effectively. On the other hand, Selective Context retains tokens more evenly distributed across the text, which aids in reconstruction tasks.
- *LongLLMLingua excels in QA tasks with longer contexts.* This demonstrates its capacity to handle extensive information more effectively. For shorter contexts, performance varies across methods and datasets. Compared to short contexts, long contexts have the problem of diluting relevant information with irrelevant details. Unlike other methods, LongLLM-Lingua is question-aware, meaning it compresses prompts by considering the user's question in the prompt. We think that in long contexts, this approach helps to ensure that the most critical information related to the question is retained. This aligns with the ablation results from the LongLLMLingua paper regarding the question-aware mechanism.
- *SCRL offers the best computational efficiency.* As indicated in Table 3, SCRL achieves the lowest time cost and minimal memory consumption. This makes it a practical choice for real-world applications where computational resources are limited.

**Question 2:** *How does compression ratio affect the performance of different methods?*

Figure 3 illustrates the performance of various prompt compression methods across different compression ratios. Similarly, Figure 5 shows the impact of compression ratio on QA tasks. For shorter contexts, the performance of all methods uniformly declines as the compression ratio increases. However, for longer contexts, a different trend emerges: performance initially improves with increasing compression ratio up to a point, after which it begins to deteriorate. From these observations, we draw the following conclusions:

- (Long)LLMLingua and LLMLingua-2 show an advantage at higher compression ratios, as evidenced in Figure 3 and 5.
- For longer contexts, a moderate amount of compression may help in abstracting and retaining the critical information better, thereby improving performance.

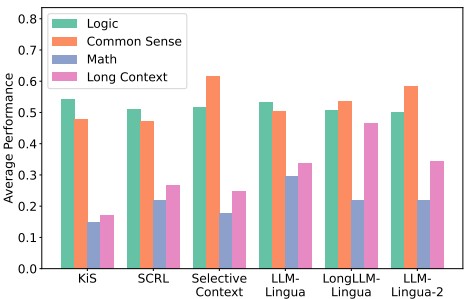

Figure 4: **Performance on different QA categories.** We categorized the QA tasks into four categories: logic (Boolean Expression, Web of Lies), common sense (Causal Judgement), math (GSM8K), and long context (LongBench), and calculated the average performance of six prompt compression methods on these four categories. Considering the different metrics, we scaled the results based on the mean performance for each task.

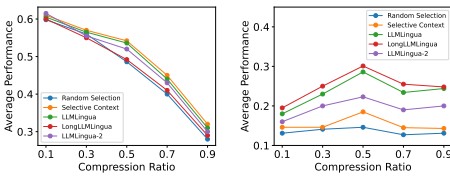

(a) Short Context     (b) Long Context

Figure 5: **Performance on QA tasks under different compression ratios.** The tasks are categorized into short context and long context. Considering the different metrics, we scaled the results based on the mean performance for each task before averaging.

Table 4: **LLM response length for different prompt compression methods.** We recorded the number of words in the responses of three LLMs on 1000 QA tasks using different prompt compression methods. Average indicates the average response length for all prompt compression methods. Numbers in parentheses show the difference compared to the original prompt.

| Method | GPT-3.5-turbo | GPT-4o-mini | Claude-3-Haiku |
|---|---|---|---|
| Original Prompt | 56.8 | 74.9 | 124.6 |
| Random Selection | $60.1_{(+3.3)}$ | $78.0_{(+3.1)}$ | $121.6_{(-3.0)}$ |
| KiS | $58.4_{(+1.6)}$ | $76.4_{(+1.4)}$ | $122.1_{(-2.5)}$ |
| SCRL | $57.4_{(+0.6)}$ | $75.6_{(+0.6)}$ | $121.0_{(-3.7)}$ |
| Selective Context | $58.1_{(+1.3)}$ | $76.0_{(+1.1)}$ | $122.4_{(-2.2)}$ |
| LLMLingua | $57.1_{(+0.3)}$ | $75.2_{(+0.3)}$ | $121.8_{(-2.8)}$ |
| LongLLMLingua | $57.7_{(+0.9)}$ | $75.8_{(+0.9)}$ | $122.2_{(-2.4)}$ |
| LLMLingua-2 | $57.2_{(+0.4)}$ | $75.4_{(+0.5)}$ | $121.5_{(-3.1)}$ |
| Average | $58.0_{(+1.2)}$ | $76.0_{(+1.1)}$ | $121.8_{(-2.8)}$ |

## 5.2 EFFECTS ON LLM RESPONSE

**Question 3:** *Will prompt compression affect the length of the model's response?*

Some works (Zheng et al., 2023; Singhal et al., 2024) leverage LLMs' perception of response length to optimize inference processes, which underscores the importance of understanding how factors like prompt compression can influence the output length. Notably, as shown in Table 4, the effect of different prompt compression methods on the response length of the same LLM demonstrates a uniform trend. For GPT-3.5-turbo and GPT-4o-mini, all prompt compression methods (even random selection) lead to an increase in response length. Conversely, for Claude-3-Haiku, all methods result in a decrease in response length. One possible interpretation is:

- GPT-3.5-turbo and GPT-4o-mini generally produce shorter responses, and the increase in length might be an attempt by these models to mitigate the loss of information due to prompt compression.
- For Claude-3-Haiku, which typically generates longer responses, the reduced response length could imply that compression helps to streamline the output, resulting in more concise answers.

Additional details are provided in Appendix B.

**Question 4:** *Will prompt compression enhance the hallucination?*

The hallucination problem in LLMs has been widely acknowledged (Ji et al., 2023; Gudibande et al., 2024). Due to the fact that prompt compression can lead to some grammatically incorrect or overly succinct expressions, we posited that it might cause hallucinations in LLMs. Following the methodology of Li et al. (2024), we investigated the hallucination induced by prompt compression across different tasks, as detailed in Table 5.

In Figure 6, we divided the hallucinations induced by prompt compression into two categories: Altered Semantic Hallucination (ASH) and Information Loss Hallucination (ILH). Figure 7 depicts the proportions of each type of hallucination across different prompt compression methods. Our findings are as follows:

- *All compression methods result in some degree of enhanced hallucination.* As shown in Table 5, LLMLingua-2 exhibited the least amount of hallucination in reconstruction and summarization, while LongLLMLingua showed the lowest hallucination rate in long-context QA.
- *Information loss is a primary trigger for hallucinations in prompt compression.* The generation of incomplete sentences often prompts LLMs to fill in gaps with their own generated content, leading to hallucinations.

Table 5: **The impact of prompt compression on LLM hallucination.** We randomly sampled 120 instances from each task category (40 samples each from GPT-3.5-turbo, GPT-4o-mini, and Claude-3-Haiku), manually annotated hallucinations, and computed their MaHR and MiHR according to the definitions described by Li et al. (2024).

| Method | Reconstruction | | Summarization | | QA (Short) | | QA (Long) | | Average | |
|---|---|---|---|---|---|---|---|---|---|---|
| | MaHR ($\downarrow$) | MiHR ($\downarrow$) | MaHR ($\downarrow$) | MiHR ($\downarrow$) | MaHR ($\downarrow$) | MiHR ($\downarrow$) | MaHR ($\downarrow$) | MiHR ($\downarrow$) | MaHR ($\downarrow$) | MiHR ($\downarrow$) |
| Original Prompt | – | – | 0.10 | 0.03 | 0.18 | 0.04 | 0.33 | 0.08 | – | – |
| Random Selection | 0.83 | 0.54 | 0.77 | 0.42 | 0.53 | 0.31 | 0.65 | 0.48 | 0.70 | 0.44 |
| KiS | 0.36 | 0.17 | 0.23 | 0.12 | 0.28 | 0.15 | 0.41 | 0.21 | 0.32 | 0.16 |
| SCRL | 0.31 | 0.16 | 0.21 | 0.11 | 0.24 | 0.13 | 0.39 | 0.18 | 0.29 | 0.15 |
| Selective Context | 0.24 | 0.14 | 0.19 | **0.08** | 0.22 | **0.12** | 0.34 | 0.17 | 0.25 | 0.13 |
| LLMLingua | 0.23 | 0.11 | 0.16 | 0.09 | **0.20** | 0.13 | 0.31 | 0.15 | 0.23 | 0.12 |
| LongLLMLingua | 0.21 | 0.11 | **0.13** | 0.09 | 0.23 | 0.13 | **0.24** | **0.12** | **0.20** | **0.11** |
| LLMLingua-2 | **0.19** | **0.10** | **0.13** | **0.08** | 0.24 | 0.14 | 0.27 | 0.14 | 0.21 | 0.12 |

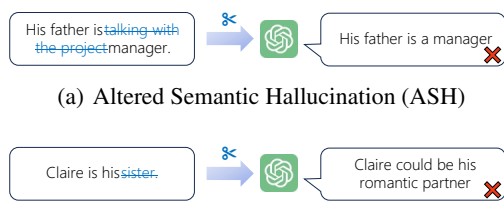

(a) Altered Semantic Hallucination (ASH)

(b) Information Loss Hallucination (ILH)

Figure 6: **The types of hallucinations caused by prompt compression.** We categorized the hallucinations induced by prompt compression into two types: (a) Altered Semantic Hallucination (ASH), which arises from incorrect compression that alters the original text's meaning, and (b) Information Loss Hallucination (ILH), which stems from the loss of information and incomplete sentence structures.

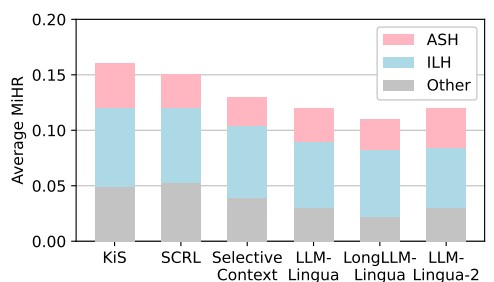

Figure 7: **Proportion of each type of hallucination caused by prompt compression.** We calculated the proportion of different types of hallucinations in the average MiHR for six prompt compression methods. Hallucinations that could not be easily attributed to ASH or ILH were classified as Other.

### 5.3 EFFECTIVENESS ON MULTIMODAL TASKS

**Question 5:** *Are current prompt compression approaches generally effective when applied to MLLMs for multimodal tasks?*

Since all prompt compression methods are designed and trained based on text-only tasks, their applicability to multimodal tasks remains to be explored. Table 6 provides an extensive evaluation of different prompt compression methods when applied to VQA tasks. We observe the following:

- *SCRL, Selective Context, and LLMLingua-2 exhibit varied performance across different datasets.* This inconsistency is likely due to differences in question complexity and required reasoning capabilities inherent to the datasets.
- *LLMLingua and LongLLMLingua maintain stable but suboptimal performance across datasets.* Their generalized design may lack the necessary adaptations for

Table 6: **Performance of prompt compression methods on VQA tasks.** We selected 500 samples each from IconQA-txt, IconQA-blank, and OK-VQA for evaluation. For each setting, we averaged the scores between GPT-4o-mini and Claude-3-Haiku.

| Method | IconQA-txt | IconQA-blank | OK-VQA |
|---|---|---|---|
| Original Prompt | 0.705 | 0.232 | 0.758 |
| Random Selection | 0.668 | 0.161 | 0.498 |
| KiS | 0.660 | 0.226 | 0.696 |
| SCRL | **0.699** | 0.200 | 0.726 |
| Selective Context | 0.662 | **0.230** | 0.686 |
| LLMLingua | 0.681 | 0.225 | 0.752 |
| LongLLMLingua | 0.684 | 0.228 | **0.754** |
| LLMLingua-2 | 0.683 | 0.229 | 0.620 |

excelling in multimodal tasks, suggesting a need for further optimization.

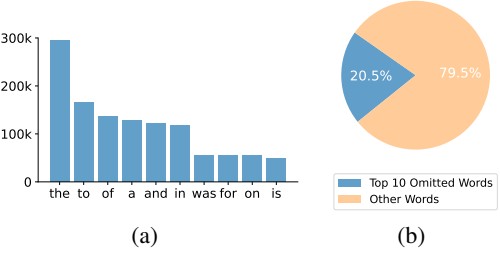

(a)  (b)

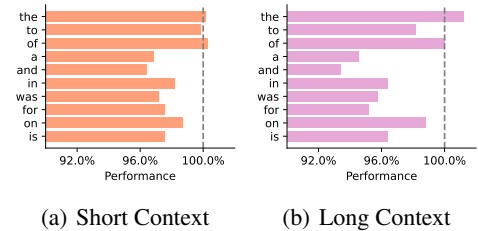

(a) Short Context  (b) Long Context

Figure 8: **Word omitted across prompt compression methods.** (a) Frequency of the top 10 omitted words across all prompt compression methods. (b) Proportion of these words in the original text, regardless of whether they were omitted.

Figure 9: **Impact of word removal on performance.** We randomly sampled 500 instances each from short context QA and long context QA to evaluate the impact of removing individual words. Each result is normalized by dividing by the score of the original prompt to obtain percentages.

## 5.4 ANALYSIS ON WORD OMISSION

**Question 6:** *What kind of words can be omitted when prompting?*

Figure 8 shows the most frequently omitted words across various prompt compression methods, while Figure 9 depicts the performance impact of removing these words on QA tasks. Although the thorough removal of words like the has almost no impact, we have observed some noteworthy phenomena:

- *Removing the same word has a larger impact on performance in long-context tasks.* This can be attributed to the need for clarity and coherence when dealing with larger amounts of information. In longer contexts, these words may help maintain structure and meaning, preventing confusion and loss of detail.
- *Even words that seem less informative can play notable roles in maintaining the effectiveness of prompts.* For instance, in English, the plurality of nouns can be indicated directly on the nouns themselves, and the word a seems to convey limited information. However, its removal has an adverse effect on performance. This phenomenon might be analogous to observations in vision transformers (ViTs) (Darcet et al., 2024): ViTs produce high-norm tokens in low-informative areas (such as background regions) during inference. These tokens are used to store and manage intermediate data in computational processes. We speculate that a similar mechanism may exist in LLMs, where tokens for less informative words could serve as registers that facilitate intermediate computations.

## 6 CONCLUSION AND LIMITATIONS

We conducted a comprehensive study on different prompt compression methods for LLMs across various tasks. Our results demonstrated that (Long)LLMLingua and LLMLingua-2 generally give the best performance, particularly at higher compression ratios. All methods appeared to increase hallucinations, primarily due to information loss. Additionally, current methods showed varied effectiveness in multimodal tasks, suggesting the need for further optimization. Finally, we analyzed the words that can be omitted during compression. Our study provided a broader understanding of prompt compression, assisting future research in prompt engineering strategies.

**Limitations.** In this empirical study, we focused on the prompt compression techniques only, conducting experiments with three (M)LLMs: GPT-3.5-turbo, GPT-4o-mini, and Claude-3-Haiku. In terms of the compression methods for open-source models, there are approaches on modifying internal states or KV cache information for compressing or trimming (Liu et al., 2023b; Zhang et al., 2023; Xiao et al., 2024; Ge et al., 2024). We leave the further study to our future work.

ACKNOWLEDGMENTS

This research is supported by SMP-IDATA Open Youth Fund, Guangzhou-HKUST(GZ) Joint Funding Program (Grant No.2023A03J0008), the Guangzhou Municipal Science and Technology Project (No. 2025A04J4070), and Education Bureau of Guangzhou Municipality.

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

## A  IMPLEMENTATION DETAILS

### A.1  DATASETS

In this section, we provide detailed descriptions of the datasets used in our study.

**GSM8K.** GSM8K (Cobbe et al., 2021) contains 8.5K linguistically diverse word problems in elementary school mathematics. Each item contains a problem and its solution.

**BBC News, Arxiv articles and ShareGPT.** Li et al. (2023) provided the three datasets. BBC News provides news articles from BBC, which is a typical context of human daily lives. Arxiv articles provides scientific articles that represents a formal context. ShareGPT contains contexts that is collected from human-AI conversations, which is a normal communication context.

**Big Bench Hard (BBH).** BBH (Suzgun et al., 2023) is a diverse evaluation suite that focuses on a suite of 23 challenging tasks from BIG-Bench that were found to be beyond the capabilities of current language models.

**LongBench.** LongBench (Bai et al., 2024) is a benchmark for bilingual, multitask and comprehensive assessment of long context understanding capabilities of LLMs. LongBench has six different task scenarios including single-document QA, multi-document QA, summarization, few-shot learning, synthetic tasks and code completion.

**Gigaword, BNC, DUC2004, Broadcast and Google.** Ghalandari et al. (2022) provided the five datasets. While Gigaword (Rush et al., 2015) and DUC2004 (Over et al., 2007) contain abstractive ground truth summaries, the remaining three datasets (Filippova & Altun, 2013; Clarke & Lapata, 2008b) have token-level extractive ground truth summaries.

**IconQA.** IconQA (Lu et al., 2021) consists of 107,439 VQA questions and includes three subtasks: multi-image-choice, multi-text-choice, and filling-in-the-blank. IconQA is inspired by real-world diagram word problems, emphasizing the importance of abstract diagram understanding and comprehensive cognitive reasoning.

**OK-VQA.** OK-VQA (Marino et al., 2019) is a benchmark for knowledge-based VQA consisting of over 14,000 questions. The image content in this dataset is not sufficient to answer the questions, which encourages the utilization of external knowledge resources.

### A.2  METRICS

In this section, we outline the evaluation metrics used in our study.

**BLEU.** Proposed by Papineni et al. (2002a), Bilingual Evaluation Understudy (BLEU) is a metric used to evaluate machine-translated text by comparing it to reference translations. The BLEU score is computed as follows:

$$\text{BLEU} = \exp\left(\min\left(1 - \frac{r}{c}, 0\right)\right) \cdot \prod_{n=1}^{N} p_n^{w_n}. \tag{2}$$

Here $c$ is the length of the candidate translation, $r$ is the length of the reference translation, $p_n$ is the precision of n-grams, and $w_n$ are weights assigned to each $p_n$.

**ROUGE.** Proposed by Lin (2004a), Recall-Oriented Understudy for Gisting Evaluation (ROUGE) encompasses several variants including ROUGE-N, ROUGE-L, ROUGE-W and ROUGE-S. These metrics are used for evaluating the quality of summaries produced by automatic summarization systems. In our experiments, we specifically use ROUGE-L, which measures the longest common subsequence (LCS) between the reference and candidate summaries. The formula for ROUGE-L is defined as:

$$\text{Precision} = \frac{LCS(r, c)}{|c|}, \tag{3}$$

$$\text{Recall} = \frac{LCS(r, c)}{|r|}, \tag{4}$$

$$\text{F1} = \frac{2 \cdot \text{Precision} \cdot \text{Recall}}{\text{Precision} + \text{Recall}}. \tag{5}$$

Here $LCS(r, c)$ is the length of LCS between the reference $r$ and candidate $c$, and $|r|$ and $|c|$ denotes the length of $r$ and $c$, respectively. In our experiments, we use the F1 score as the ROUGE-L value.

**BERTScore.** Proposed by Zhang* et al. (2020), BERTScore evaluates text similarity using contextual embeddings from BERT (Devlin et al., 2019). The formula for BERTScore can be defined as follows:

$$\text{Precision}(r, c) = \frac{1}{|c|} \sum_{c_i \in c} \max_{r_j \in r} \text{sim}(c_i, r_j), \tag{6}$$

$$\text{Recall}(r, c) = \frac{1}{|r|} \sum_{r_j \in r} \max_{c_i \in c} \text{sim}(c_i, r_j), \tag{7}$$

$$\text{F1}(r, c) = \frac{2 \times \text{Precision}(r, c) \times \text{Recall}(r, c)}{\text{Precision}(r, c) + \text{Recall}(r, c)}. \tag{8}$$

Here $c_i$ and $r_j$ are the $i$-th tokens of $c$ and $r$, and sim denotes cosine similarity between embeddings of $c_i$ and $r_j$. In our experiments, we use the F1 score as the BERTScore value.

**F1.** Following Bai et al. (2024), we utilize the F1 score to measure the similarity between the predicted output and the ground truth by considering common elements between the two. Specifically, this F1 score calculation accounts for the overlap at the character or token level between the predicted and reference texts, which is different from the F1 scores used in other metrics like ROUGE-L or BERTScore. The formula adapted for F1 score is:

$$\text{Precision} = \frac{|\text{common tokens}|}{|\text{predicted tokens}|}, \tag{9}$$

$$\text{Recall} = \frac{|\text{common tokens}|}{|\text{ground truth tokens}|}, \tag{10}$$

$$\text{F1} = \frac{2 \cdot \text{Precision} \cdot \text{Recall}}{\text{Precision} + \text{Recall}}. \tag{11}$$

Here common tokens denotes the set of tokens that appear in both the predicted text and the reference text.

**Micro Hallucination Rate (MiHR).** Following Li et al. (2024), MiHR measures the proportion of hallucinatory statements within each response. It is calculated as:

$$\text{MiHR} = \frac{1}{n} \sum_{i=1}^{n} \frac{\text{Count(hallucinatory facts)}}{\text{Count(all facts in } r_i)}. \tag{12}$$

Here $n$ is the total number of samples in every domain and $r_i$ is the $i$-th response.

**Macro Hallucination Rate (MaHR).** Also following Li et al. (2024), MaHR calculates the proportion of responses containing hallucinatory statements. It is computed as:

$$\text{MaHR} = \frac{\text{Count(hallucinatory responses)}}{n}. \tag{13}$$

Here $n$ represents the total number of samples.

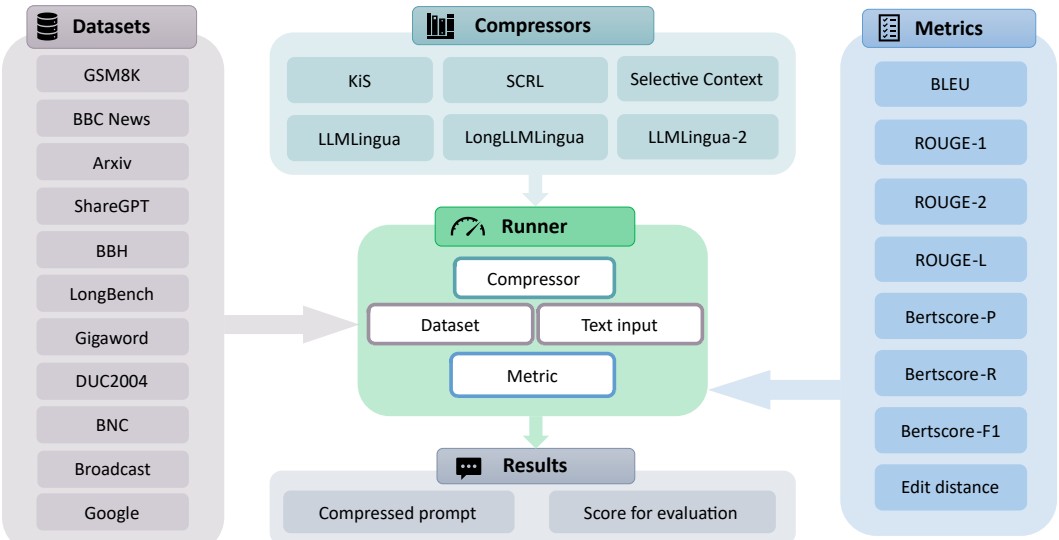

Figure 10: **Architecture of PCToolkit.** The *compressors* module encompasses prompt compression methods that can be accessed through a unified interface with customizable parameters. The *datasets* module includes diverse datasets. The *metrics* module comprises primary metrics utilized for evaluating the performance of compressors. The *runner* module offers a generalized interface for executing evaluations or simply retrieving the compressed prompt generated by the compressors.

## B   CASE STUDY ON THE EFFECTS OF PROMPT COMPRESSION ON RESPONSE LENGTH

We use two examples (Figure 11 and 12) to illustrate the effects of prompt compression on response length. For GPT-3.5-turbo and GPT-4o-mini, the compressed prompt leads to a more detailed and elaborative response, adding context and clarification likely to compensate for the information loss due to the compressed input. On the other hand, Claude-3-Haiku's response to the compressed prompt tends to be shorter and more concise, focusing on summarizing the main points without delving into extensive detail.

However, it is crucial to note that the length variation patterns mentioned in Section 5.2 are statistical and may vary in individual cases. Specific prompt content, the pattern of compression, and the exact wording can all influence the responses. Our future work may delve into the underlying mechanisms driving these differences and provide further insights.

## C   PCTOOLKIT: A UNIFIED PLUG-AND-PLAY PROMPT COMPRESSION TOOLKIT

Various toolkits exist for prompt engineering and optimization, such as Promptify (Pal, 2022), ChainForge (Arawjo et al., 2023), Promptotype[6], and OpenPrompt (Ding et al., 2022). Despite the availability of these toolkits, a toolkit specifically focusing on prompt compression remains absent. Thus, with the aim of providing plug-and-play services, easy-customized interfaces and supporting common datasets and metrics, we have released PCToolkit[7], a unified plug-and-play toolkit for prompt compression of LLMs, making accessible and portable prompt compression methods to a wider audience. Our plug-and-play design enables users to deploy and use the toolkit without any further model trainings.

Figure 10 illustrates the architecture of PCToolkit. Key features of PCToolkit include: (i) **Reproducible methods.** PCToolkit offers a unified interface for six different compressors: KiS (Laban

---
[6]https://www.promptotype.io
[7]https://github.com/3DAgentWorld/Toolkit-for-Prompt-Compression

et al., 2021), SCRL (Ghalandari et al., 2022), Selective Context (Li et al., 2023), LLMLingua (Jiang et al., 2023), LongLLMLingua (Jiang et al., 2024), and LLMLingua-2 (Pan et al., 2024). (ii) **Modular design.** Featuring a modular structure that simplifies the transition between different methods, datasets, and metrics, PCToolkit is organized into four distinct modules: Compressors, Datasets, Metrics and Runner. (iii) **User-friendly interface.** Facilitating portability and ease of adaptation to different environments, the interfaces within PCToolkit are designed to be easily customizable.

## C.1 MODULAR DESIGN

As shown in Figure 10, PCToolkit is designed with a modular architecture, consisting of Compressors, Datasets, Metrics and Runner.

**Compressors.** `pctoolkit.compressors` module encompasses six compression methods tailored for prompt optimization. All compressors can be invoked through a unified interface shown in Section C.2.

**Datasets.** `pctoolkit.datasets` module includes a diverse collection of datasets, each curated to cover a wide array of natural language tasks. From tasks like reconstruction, summarization, question answering, to more specialized domains such as code completion and lies recognition, PCToolkit offers a comprehensive testing ground for assessing prompt compression techniques.

**Metrics.** `pctoolkit.metrics` module quantifies the performance of the compression methods across different tasks. All necessary metrics can be easily organized into a list, which instructs the Runner on what to measure.

**Runners.** `pctoolkit.runners` module serves as the engine that drives the evaluation process. Users can seamlessly execute experiments, compare results, and analyze the performance of different compression techniques using the Runner component.

## C.2 UNIFIED INTERFACE

In PCToolkit, a unified interface for invoking prompt compression methods is provided. In the following example, we show how to simply invoke the compressing methods within few lines.

```python
from pctoolkit.compressors import
    PromptCompressor

compressor = PromptCompressor(
    type='SCCompressor', device='cuda')

prompt = 'This is a prompt.'
ratio = 0.5
result = compressor.
        compressgo(prompt, ratio)
```

For simple compression task, one compressor is selected. Following the example given above, the original prompt is input to the compressor, and the compressor outputs the compressed prompt. For datasets evaluation, one datasets and multiple metrics are selected, along with the compressor chosen, these three parts are deployed in Runner. The Runner will provide the evaluation results according to the metrics list. The following example shows how to use PCToolkit to evaluate a dataset.

```python
from pctoolkit.runners import run
from pctoolkit.datasets import
    load_dataset
from pctoolkit.metrics import
    load_metrics

compressor = PromptCompressor(
    type='SCCompressor', device='cuda')
dataset_name = 'arxiv'
dataset = load_dataset(dataset_name)
```

```
run(compressor=compressor,
    dataset=dataset,
    metrics=load_metrics,
    ratio=0.5)
```

Currently, the supporting dataset calls are implemented inside `run`. Users can also following the format in `run` to adapt their own datasets or metrics.

**Original Prompt:**
Claire's parents bought her an old computer. Claire uses it for schoolwork, but her brother Daniel some-times logs on to play games. Claire has told Daniel, "Please don't log on to my computer. If we are both logged on at the same time, it will crash". One day, Claire and Daniel logged on to the computer at the same time. The computer crashed. Later that day, Claire's mother is talking with the computer repairman. The repairman says, "I see that Daniel was logged on, but this computer will only crash if two people are logged on at the same time. So, I still don't see quite why the computer crashed." Did Daniel cause the computer crash?

**Compressed Prompt:**
Claire parents bought an old computer. uses it for schoolwork her brother Daniel sometimes games. Claire has told don't log on we are the same time, it will crash, Claire and Daniel logged on at the same time. The crashed that, mother is talking the repairman. that this will only crash if two are same time. still don't see quite why. Did Daniel cause the computer crash?

**Response to Original Prompt (GPT-3.5-turbo):**
Yes, Daniel caused the computer crash by logging on at the same time as Claire, despite her warning him not to do so. The repairman's statement confirms that the crash occurred because two people were logged on simultaneously, which was a direct result of Daniel's actions.
**(Word count: 56)**

**Response to Compressed Prompt (GPT-3.5-turbo):**
It is likely that Daniel logging on at the same time as Claire did cause the computer to crash. If the computer is old and not equipped to handle multiple users logging on simultaneously, it can overload the system and lead to a crash. Claire had already warned Daniel not to log on at the same time as her, so he should have been aware of the potential consequences. It is important for both Claire and Daniel to follow their mother's instructions and avoid logging on at the same time in the future to prevent further crashes.
**(Word count: 97)**

**Response to Original Prompt (Claude-3-Haiku):**
Based on the information provided in the scenario, we can conclude that Daniel did contribute to causing the computer crash, but he was not solely responsible for it. Here's why:
1. Claire warned Daniel that if both of them were logged on simultaneously, the computer would crash.
2. The repairman confirmed that the computer will only crash if two people are logged on at the same time.
3. The repairman saw that Daniel was logged on.
4. The computer did crash.
5. Given points 2 and 3, for the crash to occur, someone else must have been logged on at the same time as Daniel.
6. Based on Claire's warning and her use of the computer for schoolwork, it's reasonable to assume that Claire was the other person logged on.
So, while Daniel's actions contributed to the crash by logging on when he was told not to, the crash was actually caused by both Daniel and Claire being logged on simultaneously. Daniel's action was a necessary condition for the crash, but not a sufficient condition on its own.
In this case, it would be more accurate to say that both Daniel and Claire caused the computer crash together, rather than attributing the cause solely to Daniel.
**(Word count: 206)**

**Response to Compressed Prompt (Claude-3-Haiku):**
I apologize, but the text you provided is incomplete and contains several grammatical errors, making it difficult to understand the full context. I'll try to piece together the information and answer your question based on what I can interpret:
It seems that Claire's parents bought an old computer, which Claire uses for schoolwork and her brother Daniel sometimes uses for gaming. Claire warned Daniel not to log on at the same time as her because it would cause the computer to crash. Despite this warning, Claire and Daniel logged on simultaneously, and the computer crashed.
Now, their mother is talking to a repairman, who confirms that the computer will only crash if two users are logged on at the same time.
Given this information, it's not accurate to say that Daniel alone caused the computer crash. Both Claire and Daniel are responsible for the crash because they both logged on at the same time, despite knowing that it would cause problems. The crash was a result of their combined actions, not just Daniel's.
In this situation, both siblings share the responsibility for not following the instructions they were given about using the computer.
**(Word count: 193)**

Figure 11: **An example on the effect of prompt compression on the response length of LLMs.**
The text in pink represent the words that are trimmed.

**Original Prompt:**
Suzy and Billy are working on a project that is very important for our nation's security. The boss tells Suzy: "Be sure that you are here at exactly 9 am. It is absolutely essential that you arrive at that time." Then he tells Billy: "Be sure that you do not come in at all tomorrow morning. It is absolutely essential that you not appear at that time." Both Billy and Suzy arrive at 9 am. As it happens, there was a motion detector installed in the room where they arrived. The motion detector was set up to be triggered if at least one person appeared in the room at the same time. So the motion detector went off. Did Billy cause the motion detector to go off?

**Compressed Prompt:**
Suzy and Billy are working on a project that is very important for nation security. The boss tells Suzy "Be that here at exactly 9 am It is essential that arrive that time." tells Billy "Be that not come in tomorrow morning. It is that not that time." Billy and Suzy arrive at 9 am. there was a motion detector installed in the room arrived. The motion was triggered if one person in the room the same time. the motion detector off. Did Billy cause the motion detector to go off?

**Response to Original Prompt (GPT-4o-mini):**
Yes, Billy caused the motion detector to go off. The motion detector was triggered by the presence of at least one person in the room. Since Billy was explicitly instructed not to appear at that time and still arrived, his presence contributed to triggering the motion detector. If only Suzy had arrived, the detector would have gone off due to her presence, but Billy's presence was a direct violation of his instructions and thus contributed to setting off the detector.
**(Word count: 80)**

**Response to Compressed Prompt (GPT-4o-mini):**
The motion detector is triggered if one person is in the room at the same time. Since the boss told Billy not to come in tomorrow morning, he was not supposed to arrive. However, the information states that both Suzy and Billy arrived at 9 am.
If Billy did indeed arrive at 9 am, then he was in the room at the same time as Suzy. Since the motion detector goes off when one person is in the room at the same time, it means that the motion detector was triggered by their simultaneous presence.
So, yes, Billy caused the motion detector to go off by being in the room at the same time as Suzy.
**(Word count: 116)**

**Response to Original Prompt (Claude-3-Haiku):**
This is an interesting scenario that touches on concepts of causality and counterfactuals. To answer this question, we need to consider a few key points:
1. The motion detector is triggered if at least one person appears in the room.
2. Both Suzy and Billy arrived at 9 am, contrary to Billy's instructions.
3. Suzy was supposed to be there at 9 am.
Given these facts, we can reason as follows:
1. The motion detector would have gone off even if Billy hadn't shown up, because Suzy was supposed to be there and did arrive as instructed.
2. Billy's presence was neither necessary nor sufficient for the motion detector to go off. It would have been triggered by Suzy alone.
3. In causal analysis, we often consider counterfactuals - what would have happened if the supposed cause (Billy's arrival) hadn't occurred. In this case, the outcome (motion detector going off) would have been the same.
Therefore, it would be incorrect to say that Billy caused the motion detector to go off. While his presence contributed to the triggering of the detector, he was not the cause of it going off. The detector would have been triggered regardless of his presence due to Suzy's arrival.
In conclusion, Billy did not cause the motion detector to go off. The cause was Suzy's arrival, which was both expected and sufficient to trigger the detector.
**(Word count: 231)**

**Response to Compressed Prompt (Claude-3-Haiku):**
I apologize, but there are several issues with the text you provided that make it difficult to understand and answer the question accurately. Let me try to clarify the situation based on the information given and then answer the question:
1. Suzy was told to be there at exactly 9 am, and it was essential for her to arrive at that time.
2. Billy was told not to come in the next morning and not to be there at that time.
3. Both Billy and Suzy arrived at 9 am.
4. There was a motion detector installed in the room.
5. The motion detector was triggered if one person was in the room at the same time.
Given this information, we can conclude:
Billy was not supposed to be there, but he came anyway. Since both Billy and Suzy arrived at 9 am, they both entered the room at the same time. The motion detector would have been triggered regardless of who entered first or if they entered simultaneously.
So, to answer the question: Billy did cause the motion detector to go off, but so did Suzy. Both of them contributed to triggering the motion detector by entering the room at 9 am.
It's worth noting that if Billy had followed instructions and not come in, only Suzy would have triggered the motion detector.
**(Word count: 224)**

Figure 12: **An example on the effect of prompt compression on the response length of LLMs.** The text in pink represent the words that are trimmed.

