# OpenReview forum: "An Empirical Study on Prompt Compression for Large Language Models"
_ICLR.cc/2025/Workshop/BuildingTrust — BuildingTrust_

### Official Review · Reviewer_6NvK · 2025-02-28

**Rating:** 6
**Confidence:** 3

**Review:**

**Paper Summary**

The paper presents an empirical study on prompt compression methods for language models. It examines six distinct methods using three popular LLMs across 13 datasets. The evaluation employs different metric (BLEU, ROUGE, BERTScore for summarization tasks, downstream task performance for QA tasks) along with an analysis of hallucination rates. The study also distinguishes between long and short context settings, showing that moderate compression can improve performance on long inputs. Overall, (Long)LLMLingua and LLMLingua-2 perform best at higher compression ratios, though all methods tend to increase hallucinations, primarily due to information loss.

**Strengths**

- The study is comprehensive, evaluating six methods with three widely used models across a diverse set of tasks.
- The investigation of compression effects in both long and short contexts is well motivated, and the observation that moderate compression improves performance in long-context settings is noteworthy.

**Weaknesses**

- The significance of the differences reported in Table 4 is unclear; an average change of 1 or 2 words relative to a total of 100 words may not be meaningful.
- Some metrics lack clarity. For example, the method for averaging performance in Figures 4 and 5 is not well explained. Additionally, the histogram in Figure 8 would be more informative if the y-axis represented a relative metric rather than absolute counts. It is also unclear what fraction of the total omitted words the top 10 words represent.
- The conclusion “Removing the same word has a larger impact on performance in long-context tasks” (line 458.5)  based on? This might be the case for some of the top 10 words omitted, but might not the case for other omitted words

---

### Official Review · Reviewer_6DaB · 2025-03-02

**Rating:** 6
**Confidence:** 3

**Review:**

### Summary
This paper presents a comprehensive empirical study on prompt compression methods for LLMs. The authors evaluate six different prompt compression approaches across 13 datasets spanning various tasks including news, scientific articles, commonsense QA, math QA, long-context QA, and VQA datasets. The study goes beyond traditional performance metrics to analyze aspects such as the effect on response length, hallucination rates, effectiveness in multimodal contexts, and word omission patterns.
### Pros
1. The study performs comprehensive evaluation. It examines six compression methods across multiple dimensions and diverse datasets, providing a thorough understanding of the effectiveness of different approaches. It analyzes important aspects such as hallucination rates, response length impact, and generalizability to multimodal tasks.
2. The findings could have direct implications for practical applications of LLMs
### Cons
This paper appears to be misaligned with the workshop's focus on building trust in LLMs. The paper primarily addresses prompt compression for efficiency purposes, rather than directly tackling issues of trustworthiness.

---

### Official Review · Reviewer_hEQn · 2025-03-02
**This paper presents a comprehensive empirical study on six prompt compression methods applied to large (and multimodal) language models. The authors evaluate these techniques on a variety of tasks—including summarization, reconstruction, and question answering—across 13 datasets. They analyze not only traditional performance metrics (e.g., BLEU, ROUGE, BERTScore, accuracy) but also the impact on hallucination and response length, and provide insights into which words may be safely omitted.The paper is well-organized and exhaustive in its empirical evaluation, but several aspects warrant critical consideration.**

**Rating:** 6
**Confidence:** 4

**Review:**

Strengths

1.Comprehensive Evaluation:
 The study covers multiple prompt compression methods (including RL-based, LLM scoring-based, and LLM annotation-based techniques) and evaluates them across diverse datasets and tasks.

2.Detailed Experimental Analysis:
 The paper reports extensive results with multiple metrics (BLEU, ROUGE, BERTScore for summarization/reconstruction; accuracy and F1 for QA; and specialized hallucination metrics) and compares computational overhead.

3.Insights on Hallucination and Word Omission:
 The investigation into different types of hallucinations (Altered Semantic Hallucination vs. Information Loss Hallucination) and the analysis of word omission effects provide nuanced understanding beyond standard performance numbers.

Weaknesses

1.Limited Novelty:
 While the paper offers an excellent empirical comparison, it largely compiles and evaluates existing techniques rather than proposing fundamentally new methods. The contribution  is only incremental relative to the rapidly evolving literature on prompt engineering.

2.Methodological Justification:
 Some design choices and hyperparameter settings are not sufficiently justified. For example, while the compression ratio is set to 0.5 for certain methods, the rationale behind this choice is not deeply explored. A more in-depth discussion on parameter sensitivity could be beneficial.

3.Multimodal Task Performance:
 The extension of prompt compression techniques to multimodal settings (e.g., VQA tasks) is interesting, yet the results in this area appear less convincing. Further optimization or a more dedicated analysis might be required to strengthen these claims.

---

### Decision · Program_Chairs · 2025-03-04

Accept